# The Significance of Bayesian Pharmacokinetics in Dosing for Critically Ill Patients: A Primer for Clinicians Using Vancomycin as an Example

**DOI:** 10.3390/antibiotics12091441

**Published:** 2023-09-13

**Authors:** Faris S. Alnezary, Masaad Saeed Almutairi, Anne J. Gonzales-Luna, Abrar K. Thabit

**Affiliations:** 1Department of Clinical and Hospital Pharmacy, College of Pharmacy, Taibah University, Madinah 41477, Saudi Arabia; fnezary@taibahu.edu.sa; 2Department of Pharmacy Practice, College of Pharmacy, Qassim University, Qassim 51452, Saudi Arabia; 3Department of Pharmacy Practice and Translational Research, University of Houston College of Pharmacy, Houston, TX 77204, USA; ajgonz23@central.uh.edu; 4Department of Pharmacy Practice, Faculty of Pharmacy, King Abdulaziz University, 7027 Abdullah Al-Sulaiman Rd, Jeddah 21589, Saudi Arabia; akthabit@kau.edu.sa

**Keywords:** personalized medicine, vancomycin, intensive care units, therapeutic drug monitoring, Bayesian statistics, critically ill patient

## Abstract

Antibiotic use is becoming increasingly challenging with the emergence of multidrug-resistant organisms. Pharmacokinetic (PK) alterations result from complex pathophysiologic changes in some patient populations, particularly those with critical illness. Therefore, antibiotic dose individualization in such populations is warranted. Recently, there have been advances in dose optimization strategies to improve the utilization of existing antibiotics. Bayesian-based dosing is one of the novel approaches that could help clinicians achieve target concentrations in a greater percentage of their patients earlier during therapy. This review summarizes the advantages and disadvantages of current approaches to antibiotic dosing, with a focus on critically ill patients, and discusses the use of Bayesian methods to optimize vancomycin dosing. The Bayesian method of antibiotic dosing was developed to provide more precise predictions of drug concentrations and target achievement early in therapy. It has benefits such as the incorporation of personalized PK/PD parameters, improved predictive abilities, and improved patient outcomes. Recent vancomycin dosing guidelines emphasize the importance of using the Bayesian method. The Bayesian method is able to achieve appropriate antibiotic dosing prior to the patient reaching the steady state, allowing the patient to receive the right drug at the right dose earlier in therapy.

## 1. Introduction

The global utilization of antimicrobial agents has contributed to the exponential growth of bacterial resistance on a worldwide basis [1,2,3]. Despite the approval of several antibiotics in recent years targeting the treatment of drug-resistant bacteria, difficult-to-treat infections have become more common, and the pace of new approvals has not kept up with the development of resistance, creating a challenge for clinicians [4,5,6]. As a result, there has been increasing interest in improving the utilization of current antibiotics through stewardship efforts and the optimization of dosing.

Recent studies focusing on critically ill populations brought to light several challenges facing clinicians. Altered pathophysiology in the pharmacokinetics (PKs) and pharmacodynamics (PDs) of antibiotics is one of the main causes of suboptimal antibiotic exposure. These PK/PD changes are difficult to track among different patient populations, patients in the same population, and even individual patients over time [7]. Recent studies suggest that the current dose adjustment strategies do not account for these dosing challenges [8]. Indeed, standard antimicrobial dosing administered to severely ill patient populations often results in suboptimal antibiotic exposures, emphasizing the need for dose optimization to ensure positive patient outcomes [9,10]. This can be achieved in part through individualized antibiotic dosing based on patient-specific PK/PD changes [9,11]. However, in clinical practice, it is difficult to know to what extent and how doses should be modified.

One of the most common antibiotics administered to critically ill patients is vancomycin, a glycopeptide antibiotic, which has been the primary agent used in hospitalized patients with methicillin-resistant Staphylococcus aureus (MRSA) infections since the 1950s. An updated consensus guideline for monitoring vancomycin in the treatment of serious MRSA infections was recently published by the American Society of Health-System Pharmacists (ASHP), the Infectious Diseases Society of America (IDSA), the Pediatric Infectious Diseases Society (PIDS), and the Society of Infectious Diseases Pharmacists (SIDP) [12]. The revised 2020 guidelines emphasize the importance of using Bayesian software programs and recommend their use over traditional PK calculations to determine the area under the curve (AUC) of drug concentration estimates. 

This narrative review aims to summarize the challenges in current antibiotic dosing regimens in critically ill patients and discuss a newly recommended Bayesian approach to optimize vancomycin dosing in this population. We hope this review serves as a primer for clinicians in selecting appropriate antibiotic doses in critically ill populations, as well as other patient populations with altered PK/PD.

## 2. Altered PKs in Critical Illness

Critically ill patients and patients with trauma, burns, febrile neutropenia, major surgery, organ dysfunction, obesity, or cystic fibrosis often have drastic changes in their physiology [10,13,14,15]. These significant physiological changes are accompanied or caused by a state of systemic inflammation and may result in end-organ system failure, requiring renal replacement therapy or extracorporeal membrane oxygenation [16,17,18]. Such organ-saving technologies often present their own additional PK challenges, in addition to those inherent to the predisposing disease state. 

Body fluid dynamics may have a significant impact on the PKs of many antibiotics. Most critically ill patients have altered body water volumes, resulting from dynamic cardiovascular and renal function or substantial fluid resuscitation [19,20]. Fluids accumulate in these expanded extracellular spaces, diluting antibiotic concentrations. Furthermore, increases in cardiac output lead to increased renal clearance and faster drug elimination [21].

### 2.1. Changes in Volume of Distribution (Vd)

There are several factors that have the potential to alter the volume of distribution (Vd) in the context of critical disease. Changes in the volume of distribution (Vd) primarily occur due to fluid administration during the initial stage of shock therapy [22]. The expansion of the extracellular fluid (ECF) volume occurring with systemic inflammation results in an increased Vd of hydrophilic antibiotics. In patients with hypoalbuminemia, the plasma oncotic pressure is reduced, which causes an increase in ECF volume, and hence Vd. Therefore, using highly protein-bound drugs, such as ceftriaxone (83–95% protein-bound), in critically ill patients with hypoalbuminemia could increase Vd up to 90% [23]. Vd may also expand due to fluid accumulations in patients with ascites, pleural effusions, or mediastinitis [24,25,26]. These significant Vd changes primarily impact antibiotics that are dependent on high concentrations for efficacy as increases in Vd reduce peak concentrations. Therefore, dose optimization in critically ill patients is needed to overcome the changes in Vd [27].

### 2.2. Changes in Drug Elimination

Critical illness and shock may lead to acute kidney injury (AKI). Consequently, AKI often results in supratherapeutic antibiotic concentrations depending on drug-specific metabolic profile. Drugs that are renally cleared, such as cefepime, accumulate during AKI and may cause adverse events if the dose is not adjusted accordingly [28,29,30]. However, alternative, nonrenal clearance mechanisms are often upregulated during AKI and allow drugs such as flucloxacillin and ciprofloxacin to be efficiently cleared even during AKI. However, when multiorgan failure occurs, both renal and nonrenal clearances are impaired, resulting in supratherapeutic antibiotic concentrations if doses are not adjusted appropriately [31,32,33,34]. Variability in drug clearance among AKI patients with extracorporeal therapies, such as renal replacement therapy, also exists [35,36,37]. Another issue complicating the management of patients with AKI is the development of tubular impairment, which affects both secretion and reabsorption. Impairment in tubular secretion may decrease the clearance of β-lactam antibiotics but may augment it for other antimicrobials, such as fluconazole, which encounters significant reabsorption, necessitating higher doses [35,38]. These clearance variations indicate the complexity of drug dosing in patients with AKI.

In contrast, several factors associated with critical illness may increase renal clearance. As described previously, elevated cardiac output increases renal blood flow and eventually increases glomerular filtration. This process is more common in patients with critical illness due to sepsis, trauma, burn injury, acute leukemia, and major surgery [39,40,41]. Fluid therapy and vasoactive drugs may also influence renal blood flow and glomerular filtration rate, leading to enhanced renal clearance of antibiotics (augmented renal clearance (ARC)) [21]. ARC, defined as a creatinine clearance (CrCl) >130 mL/min/1.73 m^2^, occurs in some critically ill and >85% of trauma patients [39,40,42]. ARC increases antibiotic elimination and is more likely to impact antibiotics that are time-dependent for efficacy.

Furthermore, changes in serum creatinine (SCr), which is the most commonly used clinical surrogate in determining kidney function, often lag compared with actual renal function, especially in critically ill patients. This creates a challenge for clinicians in adjusting antimicrobial doses even when seemingly appropriate [43]. Clinicians adjusting drug doses based on SCr may decrease the dose of renally cleared drugs despite improving renal function, resulting in subtherapeutic concentrations in some patients. As an alternative, clinicians are recommended to use cystatin C concentration as a surrogate marker for kidney function since it changes faster with changes in kidney function and has been found to be significantly associated with AKI incidence estimation and ICU mortality compared with SCr concentration in critically ill patients with sepsis [44,45,46]. These findings were evident even in critically ill elderly patients as well as patients receiving continuous renal replacement therapy [47,48]. When it comes to vancomycin clearance estimation and its association with kidney function, a meta-analysis of 26 studies by Tahir et al. revealed that calculations of the estimated glomerular function rate (eGFR) based on cystatin C were better in predicting the clearance of vancomycin than eGFR calculation using SCr values [49]. In this study, the authors aimed to evaluate the degree of bias in vancomycin clearance prediction using eGFR equations considering either cystatin C or SCr. The latter was significantly associated with biased results in predicting vancomycin clearance, where the mean prediction error of the SCr-based eGFR estimation was 27.62 mL/min (95% CI, 8.68 to 46.56). On the other hand, the mean prediction error of cystatin C-based eGFR was only 4.38 mL/min (95% CI, −29.43 to 38.18), indicating an unbiased estimation. Furthermore, the degree of precision of the cystatin C-based eGFR equation in predicting vancomycin clearance was higher than that of the SCr-based eGFR equation based on the root-mean-squared error results (28.96 vs. 61.56, respectively). The results supporting cystatin C utilization persisted after excluding the studies performed on neonates, which indicates that it is applicable in the adult population. Interestingly, when both approaches of eGFR calculation were compared in predicting vancomycin clearance using Bayesian PK and two-stage first-order kinetics, the cystatin C-based estimation of eGFR remained superior to SCr-based eGFR estimation, with a mean prediction error of 3.20 mL/min (95% CI, −22.90 to 29.31) and 4.89 mL/min (95% CI, −38.07 to 47.85) for Bayesian PK and two-stage approaches, respectively. The findings from this meta-analysis further support the use of cystatin C-based estimation of kidney function in PK calculations for dose adjustment in critically ill patients as a substitute for SCr-based calculations. As such, developers of Bayesian PK programs are recommended to incorporate the cystatin C equation in their software and to add a question if the patient for whom the dose is to be calculated is critically ill. The cystatin C-based equation should be used in place of the SCr-based equation for kidney function estimation only in this patient population.

## 3. Antibiotic Dosing Approaches and Variability

Antibiotic dosing approaches vary greatly depending on a variety of factors, including patient population, resources, settings, and antibiotic properties. Five such strategies are discussed here and summarized in Table 1.

### 3.1. Fixed Dosing

Fixed dosing regimens are commonly used for drugs with a wide therapeutic index (e.g., omeprazole 20 mg daily and amoxicillin 500 mg every 8 h). However, drugs with a narrow therapeutic index (NTI) or with variable intra- and inter-patient responses often require a different approach. Vancomycin has an NTI and therefore has historically been dosed based on trough concentrations. However, PK modeling has demonstrated that around half of vancomycin trough concentrations were outside the therapeutic range (10–20 mg/L), despite the use of standardized dosing guidelines [50].

### 3.2. Covariate-Based Dosing

The principle of “one dose fits all” has been replaced for some drugs by covariate-based dosing, which utilizes patient covariates (patient-specific variables), such as weight (for mg/kg dosing) and renal function, to decrease the variability in drug response between patients [51]. For several NTI drugs, including some antibiotics, anticoagulants, and chemotherapeutic agents, covariate-based dosing has become the standard practice, as reflected in published guidelines [12,52,53]. Nevertheless, there are some limitations to this method as dose adjustment is carried out according to a single point estimate of the covariate, which is not a strong indicator of PK variability. For example, many dosing protocols may recommend that a dose should be reduced by 50% if the patient is older than 65 years of age or has a CrCl <50 mL/min. Unfortunately, this type of threshold-based dosing recommendation does not accurately correlate with fluctuations in physiological functions. Furthermore, the suggested dosing recommendations are often broad and do not take into account patient-specific factors within populations, which increases the chance for variability between patients and thus impacts outcomes. For instance, for clinicians wanting to calculate a loading dose of 20–30 mg/kg, it would be reasonable to select a dose ranging from 1400 mg to 2100 mg for a 70 kg patient. This has been shown to result in poor patient outcomes with some antibiotics due to sub- or supratherapeutic drug concentrations [54,55]. As a result, patient responses are often unpredictable when covariate dosing is used.

### 3.3. Dosing Based on Therapeutic Drug Monitoring (TDM)

Therapeutic drug monitoring (TDM) has therefore been developed and is used with covariate-based dosing for some drugs with an NTI or that have significant PK variability to optimize efficacy and prevent adverse effects. TDM refers to the practice of measuring individual drug plasma concentrations in order to maintain concentrations within a predetermined therapeutic range [56]. TDM is based on an assumed relationship between drug plasma concentration and therapeutic effect. Accordingly, the benefit of TDM may be greatest when high concentrations result in severe adverse events or low concentrations result in therapeutic failure. The main challenge with TDM is the required precision when measuring levels, as the blood sample needs to be drawn within a specific and prespecified time window for accurate interpretation. Additionally, TDM requires drug-specific validated and accessible assays with clearly defined reference ranges.

Although TDM allows for more informed dose adjustments, exclusive reliance on TDM for dose adjustments has several limitations. TDM shares the same drawback as fixed and covariate-based dosing: Their development is based on the “average” patient response, which may not be accurate to address variability between patients [51]. TDM may also result in a “trial and error period” in which clinicians must wait for a given serum level to return prior to adjusting a dose and is often associated with delays in achieving target concentrations. TDM-based dosing also carries increased healthcare costs associated with ordering extra serum concentrations needed to optimize the dose following or preceding dose adjustments [57,58]. Two other methods of dose individualization have emerged to target specific PK parameters without limitations faced by TDM and many other conventional dosing strategies: population methods using nomograms and Bayesian estimation procedures [59].

### 3.4. Nomogram-Based Dosing

In the population method, antibiotic dosage can be determined based on population PK parameters without using patients’ individual PK results [48]. Population models are developed using Monte Carlo simulations, which simulate the PK parameter results of a few patients on thousands of patients. The Hartford Nomogram for aminoglycosides is a typical example, which uses the estimates of PK parameters, such as Vd, to provide dose recommendations for aminoglycosides. Nomograms provide easy interpretation as limited PK knowledge is required, and the implementation and use of nomograms do not require advanced technology. Additionally, nomograms only require minimal patient information. Nevertheless, nomograms have some noteworthy limitations. First, nomograms have only been developed for a limited number of antibiotics. Second, this method does not take into account the changes in PK parameters for special populations, including patients with burns, pregnancy, ascites, or dialysis [60]. Additionally, previous studies have found discrepancies in the dose recommendations obtained from the population-based nomogram methods [61,62]. As a result, nomograms are no longer being recommended for vancomycin dosing [12].

### 3.5. Bayesian Dosing

The Bayesian method was developed to overcome the issue of variability between patients and provide more precise predictions of drug concentrations and target achievement early in therapy [63]. The Bayesian method utilizes information on the drug’s behavior in previous patient populations, including drug clearance and Vd (a priori), in conjunction with the current patient-specific information, including weight, measured drug concentration(s), and PK parameters, calculated based on the administered dosing regimen (a posteriori), to develop a precise dose recommendation for a specific patient [59,64]. Figure 1 demonstrates the development process for individualized dose recommendations using a Bayesian PK method. The software combines these data and analyzes them using multiple mathematical equations to calculate an individualized dose for each patient to attain the PK/PD target [65].

The Bayesian method has several advantages over traditional dosing (Table 1). First, individualized PK/PD parameters can be calculated based only on a single drug serum concentration, which could minimize the number of required blood draws for serum drug concentration testing and thereby improve patient comfort [63,66]. Second, Bayesian dosing strategies have demonstrated greater predictive abilities and precision to achieve targeted concentrations than fixed dosing strategies [67,68]. Third, Bayesian programs have been shown to improve patient care by minimizing drug toxicity and maximizing drug efficacy [63,66]. Notably, individualized Bayesian dosing is more beneficial for some patient populations, such as patients with altered physiologic functions [69]. Clinicians should employ clinical knowledge and judgment while using Bayesian software programs, as different Bayesian programs can provide an individual patient with substantially different dose recommendations [70]. These differences could be due to assumptions in the PK distribution in the models, which requires routine assessment.

## 4. Application of the Bayesian method in Vancomycin Dosing

In the previous 2009 guideline, the target dosage of vancomycin was recommended to be a serum trough concentration of 10–20 mg/L, which was considered a surrogate marker for an AUC to minimum inhibitory concentration (AUC/MIC) ratio of 400–600 [71]. Emerging data since then have demonstrated that trough concentrations may not be good surrogate markers for AUC values; therefore, the 2020 update of the guidelines recommended directly targeting an AUC/MIC ratio of 400–600 based on an MRSA MIC of 1 mg/L. This represents a major shift in clinical practice, and recommendations are provided for the implementation and monitoring of vancomycin, as well as for dosing in special populations.

Calculating AUC/MIC can be performed using either first-order PK or Bayesian PK software. The first involves the collection of two serum concentrations at the steady state (a peak after the third dose and a trough before the fourth dose) and then calculating the AUC using a mathematical equation. Alternatively, Bayesian programs require only one concentration collected within 24–48 h of starting therapy instead of waiting until achieving the steady state. As a result, these programs are able to calculate and predict AUC/MIC earlier during a patient’s treatment course. Bayesian programs also offer the advantage of increased precision since many allow for the optional addition of patient-specific variables, such as age, weight, height, and serum creatinine level. Indeed, a recent study on 116 patients found that Bayesian-guided AUC estimation resulted in faster results that were 20–25% more accurate than the results obtained using first-order PK equation-based AUC calculation [72]. Similarly, a study including 124 patients found that AUC estimation using Bayesian PK outperformed first-order PK calculation, and vancomycin concentrations were deemed usable for AUC calculations in 88.2% vs. 48.3% using each method, respectively [73]. Moreover, that study also found that integrating the Bayesian software with the electronic medical system of the hospital resulted in a shorter time to dose recommendation compared with manual calculation or the manual use of calculators to estimate the dose using the first-order PK equation. Similar findings supporting the accuracy of AUC predictions using Bayesian PK have also been reported in the pediatric population [74]. Additionally, it is worth noting that most Bayesian PK programs have user-friendly interfaces, allowing for easy implementation in clinical practice settings and minimal training. In critically ill patients, the frequency of vancomycin monitoring is left to the judgment of the clinician and can be performed daily or weekly based on the hemodynamic stability of the patient and the status of stable or fluctuating kidney function. In the case of rapidly changing kidney function in critically ill patients, the use of cystatin C-based estimation of kidney function is recommended since it correlates better with AKI incidence and vancomycin clearance prediction than SCr-based estimation [44,49]. Therefore, dose calculation based on kidney function that is estimated using cystatin C concentrations should be incorporated in Bayesian PK programs to be utilized by clinicians caring for critically ill patients in ICUs.

Considered a common alternative to intermittent dosing, another vancomycin dosing strategy used in ICUs is continuous infusion, which involves a continuous IV infusion of vancomycin over 24 h. Such an approach was associated with a lower risk of nephrotoxicity and better economic outcomes, as it required less monitoring than intermittent infusion [12,75]. The 2020 vancomycin guidelines provide specific recommendations on this dosing approach, according to which a loading dose of 15–20 mg/kg would be given first, followed by a continuously infused daily maintenance dose of 30–60 mg/kg [12]. The target steady-state vancomycin concentration with this approach is 20–25 mg/L, which reflects an AUC/MIC of 480–600 (based on an MRSA MIC of 1 mg/mL). The calculation of AUC/MIC can be performed by multiplying the vancomycin concentration by 24 to estimate the AUC_0–24_ (i.e., AUC over 24 h). The steady-state concentration is recommended to be taken 24 h after the initiation of continuous infusion. If dose adjustment is needed, it should be implemented in increments of 250–500 mg. While the TDM of vancomycin with this dosing method can be carried out by simply drawing a level and multiplying it by 24 (i.e., without the utilization of the Bayesian method), utilizing the Bayesian method for vancomycin dose adjustment with continuous infusion was significantly associated with better rate of target attainment (a vancomycin concentration of 20–25 mg/L) than the standard TDM approach (85% vs. 57%; *p* = 0.007) [76]. However, it should be noted that the study revealing this finding was conducted on critically ill children (aged 3 months to 17 years). Hence, a similar study on adults is warranted to confirm these results in this population since drug PKs largely differ between the two different populations.

## 5. Conclusions

Growing evidence indicates that antibiotic dosing must be tailored based on patient-specific PK/PD changes to provide positive clinical outcomes. Patients with critical illness may experience a significant impact on the PKs of many antibiotics. A number of factors may affect antibiotic distribution in critical illness. There is an increased volume in the ECF with systemic inflammation, decreased plasma oncotic pressure, and increased antibiotic Vd. Critical illness and shock may cause AKI, which can lead to subtherapeutic or supratherapeutic antibiotic concentrations depending on the drug-specific metabolic profile.

TDM is used to optimize efficacy and prevent adverse effects but requires precision to address variability between patients. It has other limitations as well, such as trial and error and delays in achieving target concentrations. The Bayesian method was developed to overcome these issues and provide more precise predictions of drug concentrations and target achievement early in therapy, with advantages such as individualized PK/PD parameters, greater prediction abilities, and improved patient care.

Since recent research has emphasized the significance of implementing the Bayesian method for vancomycin dosing, it would be prudent to investigate the role of the Bayesian method in optimizing the dosage of other antibiotics, particularly those that require TDM. This will help achieve appropriate antibiotic dosing prior to the patient reaching the steady state, allowing patients to receive the right drug at the right dose earlier in therapy. The use of Bayesian programs will aid in the standardization of dosing across a wide range of institutions and health systems. Furthermore, having Bayesian programs available at the point of care will save clinicians time while also improving patient safety. Future research should be conducted to demonstrate the wide range of benefits of the Bayesian approach.

## Figures and Tables

**Figure 1 antibiotics-12-01441-f001:**
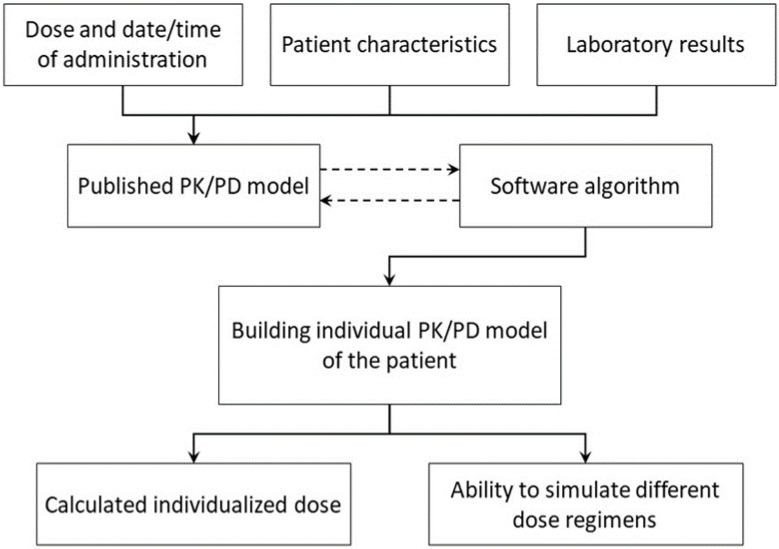
The development process for individualized dose recommendations using a Bayesian PK method.

**Table 1 antibiotics-12-01441-t001:** Summary of advantages and disadvantages of different dosing methods.

Dosing Method	Advantages	Disadvantages
Fixed dosing	Accessible and easy to use	Not accurate (no drug PK variability taken into account between subjects)
Covariate dosing	Cheap, accessible, and easy to use	Requires determination of covariate Requires blood concentrations to be withdrawn at certain times Several body size descriptors, such as ideal body weight, need to be evaluated
Therapeutic drug monitoring	Supports dose individualization Improves the safety of some drugs	Requires accurate assays Requires some clinical experience Can be only used for drugs with a predefined therapeutic range Requires blood concentrations to be withdrawn at certain times
Nomogram	Provides simple interpretation Requires limited PK expertise No advanced technology is required for implementation and use	Only a small number of antibiotics are used Does not take into consideration changes in PK parameters in special populations Variations in dosage recommendations
Bayesian dosing	Can predict drug concentrations May use single blood concentration in some cases Provides more accurate prediction with more patient information Provides flexibility for blood sampling time	Requires software (cannot be performed by hand) Expensive (depends on software) Requires some training (basic or advanced, depending on the software) Incorrect drug selection leads to inappropriate dose recommendation Requires PK models for the population of interest

PK, pharmacokinetic.

## Data Availability

Data are available form the authors upon request.

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
