# Peer review of "The Significance of Bayesian Pharmacokinetics in Dosing for Critically Ill Patients: A Primer for Clinicians Using Vancomycin as an Example"

_antibiotics, 2023, doi:10.3390/antibiotics12091441_

Round 1

Reviewer 1 Report

The title of the paper is “The Role of Bayesian Pharmacokinetics in Vancomycin Dosing in Critically Ill Patients: A Primer for Clinicians”. However the paper mainly compared the advantages and disadvantages of current approaches of the antibiotic dosing as shown in Table 1. Discussions on the use of Bayesian methods to optimize vancomycin dosing are a little bit weak.

Comments:

Vancomycin has be used in clinic for over 50 years in the treatment of serious Gram-positive bacterial infections. It is still used in clinic. Dosing regimens are critically related to the outcomes. The current review picked an interesting topic. However, in the 65-cited papers, only 9 papers were within 5 years. The text content related to Vancomycin is insufficient. Please include more recent papers on the clinical application of Vancomycin in the critically ill patients and discuss the use of Bayesian methods on the optimization of vacomycin dosing.

Please use the same fomat for the references. 

Author Response

1- Vancomycin has been used in clinics for over 50 years in the treatment of serious Gram-positive bacterial infections. It is still used in clinics. Dosing regimens are critically related to the outcomes. The current review picked an interesting topic. However, of the 65-cited papers, only 9 papers were within 5 years.

Response:

1- With regard to the use of papers published beyond a five-year timeframe, it is our opinion that despite the significance of this subject matter for the outcomes of our patients, it has been insufficiently addressed in the recently published literature. Hence, it is imperative to reiterate the key concepts described in the research to clinicians working in the field.

2- The text content related to Vancomycin is insufficient. Please include more recent papers on the clinical application of Vancomycin in critically ill patients and discuss the use of Bayesian methods on the optimization of vacomycin dosing.

Response:

2- With regards to the use of Bayesian method on the optimization of vancomycin dosing, we see that the revised guidelines for vancomycin highlight the significance of employing the Bayesian strategy, as it allows for the collection of vancomycin concentrations within the initial 24 to 48 hours, instead of relying solely on steady-state concentration. This technique aids in informing subsequent dosing decisions. Furthermore, Bayesian dosing programs offer novel treatment strategies in order to rapidly attain targeted concentrations within the initial 24 to 48 hours in critically ill individuals. However, it is important to observe that, to our knowledge, there are no published studies on the outcomes of applying the Bayesian method to special populations, such as critically ill patients. Therefore, we will express the need to do further research in order to examine the specific outcomes for this particular population.

Reviewer 2 Report

Thank you for the opportunity to review the manuscript entitled “The Role of Bayesian Pharmacokinetics in Vancomycin Dosing in Critically Ill Patients: A Primer for Clinicians”.

In this paper, the authors reviewed the existing literature on the advantages and disadvantages of current approaches to antibiotic dosing, with a focus on Bayesian methods to optimize vancomycin dosing in critically ill patients. They emphasize how the Bayesian method of antibiotic dosing could provide more precise predictions of drug concentrations and target achievement and could improve patient outcomes. 

The authors are to be commended for their efforts to provide valuable information on this pertinent question. This paper is interesting, the manuscript is appropriately organized, but overall, the review lacks maturity and hindsight on what Bayesian PK can bring to patients today. Notably, no large-scale study has shown the benefit of using Bayesian PK to decrease mortality in the ICU. It is important to underline that ICU patient prognosis is not only linked to the correct use of antibiotics, but also to the way patients are ventilated, hemodynamics is controlled, and life organ support are performed... I'll give you some comments to try to improve the paper, but I think it should be reworked with a clinician who has experience in the field. 

·      Introduction 

a)     I totally disagree with the first sentence of the introduction. It's not the delays in the development of new antibiotics over several decades that have facilitated the emergence of bacterial resistance, but the consumption of antibiotics and the ability of bacteria to adapt by resisting. This sentence needs to be rewritten, as does the 1st paragraph of the introduction.

·      Altered PK in critical illness 

a)     Changes in volume of distribution (Vd) is also and mainly caused by fluid therapy in the early phase of shock therapy. Please change the text and add citations. 

b)    “These significant Vd changes primarily impact antibiotics that are dependent on high concentrations for efficacy as increases in Vd reduce peak concentrations.” May I ask the authors which antibiotic PK is not altered by a increase or a decrease in their Vd? Qualify this sentence.

c)     “Consequently, AKI often results in subtherapeutic or supratherapeutic antibiotic concentrations depending on drug-specific metabolic profile.” it's mostly supratherapeutic concentrations depending on the specific metabolic profile of the drug subtherapeutic antibiotic concentrations are mostly caused by doctors lowering doses without knowing the drug's metabolism (such as ceftriaxone and others)... what do you think?

d)    “ARC, defined as a creatinine clearance (CrCl) >130 mL/min/1.73 m2, occurs in >50% of critically ill”. Depends on the type of intensive care unit... but overall it's not true, it's much less frequent, especially in medical intensive care units.

e)     Clinicians adjusting drug doses based on SCr may decrease the dose of renally cleared drugs despite improving renal function resulting in subtherapeutic concentrations in some patients. True. But this parameter which is not a good surrogate as correctly pointed by the authors is almost ever incorporated in the PK models… 

f)     For instance, for clinicians wanting to calculate ‘a loading dose of 20–30 mg/kg’, it would be reasonable to select a dose ranging from 1400 mg to 2100 mg for a 70 kg patient. This has shown to result in poor patient outcomes with some antibiotics due to sub- or supratherapeutic drug concentrations [44,45]. In my opinion, these are not the recommendations worldwide…

·      Application of the Bayesian method in vancomycin dosing. Some physicians prefer to give vancomycin by continuous infusion after a loading dose, especially in ICU setting. Please add a statement on this and discuss the benefit of this dosing regimen. Then, the use of Bayesian method to improve target attainment in patients treated with continuous infusion of vancomycin.

·      Conclusion: 

a)     “It would be prudent to use the Bayesian approach to achieve appropriate antibiotic dosing prior to the patient reaching the steady state, allowing patients to receive the right drug at the right dose earlier in therapy”. Are the authors certain that the use of Bayesian PK approach is not associated with arm in critical ills and associated with some benefit? What is the evidence? 

b)    The title of the study focus on vancomycin, but the main text and especially the conclusion deal with Bayesian PK… please rethink the title, the purpose and the conclusion +++ of this paper ! 

 One more time, thank you for the opportunity to review your manuscript.

Author Response

Point 1: a) I totally disagree with the first sentence of the introduction. It's not the delays in the development of new antibiotics over several decades that have facilitated the emergence of bacterial resistance, but the consumption of antibiotics and the ability of bacteria to adapt by resisting. This sentence needs to be rewritten, as does the 1st paragraph of the introduction.

Response: We appreciate your comment, and in response, we carefully reviewed the words and supplemented the revised sentence with relevant evidence to substantiate its claims.

Point 2: Altered PK in critical illness a) Changes in volume of distribution (Vd) is also and mainly caused by fluid therapy in the early phase of shock therapy. Please change the text and add citations.  

Response: The sentence was changed, and a new sentence was added to line 86.

Point 3: b)    “These significant Vd changes primarily impact antibiotics that are dependent on high concentrations for efficacy as increases in Vd reduce peak concentrations.” May I ask the authors which antibiotic PK is not altered by a increase or a decrease in their Vd? Qualify this sentence.

Response: We added a sentence to make it clear. Antibiotics PKs in critically ill patients, in this case Vd, may be altered by an increase or decrease in total body water and intravascular volume. It is crucial to understand these changes in Vd to optimize the doses for critically ill patients

Point 4: c)     “Consequently, AKI often results in subtherapeutic or supratherapeutic antibiotic concentrations depending on drug-specific metabolic profile.” it's mostly supratherapeutic concentrations depending on the specific metabolic profile of the drug subtherapeutic antibiotic concentrations are mostly caused by doctors lowering doses without knowing the drug's metabolism (such as ceftriaxone and others)... what do you think?

Response: In order avoid any confusion, the term "subtherapeutic" was removed.

Point 5: d)    “ARC, defined as a creatinine clearance (CrCl) >130 mL/min/1.73 m2, occurs in >50% of critically ill”. Depends on the type of intensive care unit... but overall it's not true, it's much less frequent, especially in medical intensive care units.

Response: To improve the accuracy of the statement, the expression ">50%" was replaced with the term "some."

Point 6: e)     Clinicians adjusting drug doses based on SCr may decrease the dose of renally cleared drugs despite improving renal function resulting in subtherapeutic concentrations in some patients. True. But this parameter which is not a good surrogate as correctly pointed by the authors is almost ever incorporated in the PK models.

Response: We agree on this point. As SCr is not a good surrogate of kidney function in critically ill patients, many clinicians prefer to use cystatin C as an alternative parameter as it changes faster than SCr in response to changes in kidney function. A recommendation for Bayesian software developers to include an option to add cystatin C values in the PK calculations was added on line 132 with a justification on why cystatin C is preferred over SCr in this patient population.

Point 7: f)   For instance, for clinicians wanting to calculate ‘a loading dose of 20–30 mg/kg’, it would be reasonable to select a dose ranging from 1400 mg to 2100 mg for a 70 kg patient. This has shown to result in poor patient outcomes with some antibiotics due to sub- or supratherapeutic drug concentrations [44,45]. In my opinion, these are not the recommendations worldwide

Response: According to the latest vancomycin guidelines, it is recommended to administer a loading dose of 20-35 mg/kg for intermittent administration of vancomycin in critically ill patients with suspected or confirmed serious MRSA infections. However, there is considerable variability among clinicians regarding the specific dosage to be administered. For instance, for a patient weighing 70 kg, the dosage can range from approximately 1400 mg (or around 1500 mg) to 2450 mg (or around 2500 mg), resulting in a difference of 1 gram between two clinicians.

Point 8: Application of the Bayesian method in vancomycin dosing. Some physicians prefer to give vancomycin by continuous infusion after a loading dose, especially in ICU setting. Please add a statement on this and discuss the benefit of this dosing regimen. Then, the use of Bayesian method to improve target attainment in patients treated with continuous infusion of vancomycin.

Response: Additional discussion on continuous infusion of vancomycin and its benefits to critically ill patients was added to line 295.

Point 9: Conclusion: a)     “It would be prudent to use the Bayesian approach to achieve appropriate antibiotic dosing prior to the patient reaching the steady state, allowing patients to receive the right drug at the right dose earlier in therapy”. Are the authors certain that the use of Bayesian PK approach is not associated with arm in critical ills and associated with some benefit? What is the evidence?

Response: We attempted to provide a similar explanation to that which was presented in the most recent vancomycin recommendations. To far, there has been a lack of literature pertaining to the outcomes associated with the use of Bayesian dosing in critically ill patients. Nevertheless, our study proposes a theoretical advantage that warrants consideration.

Point 10: b)    The title of the study focus on vancomycin, but the main text and especially the conclusion deal with Bayesian PK… please rethink the title, the purpose and the conclusion +++ of this paper !  

Response: The title has been revised to provide a more precise and accurate description, as recommended by the reviewer.

Round 2

Reviewer 1 Report

Typos:

Line 123:(CrCl) >130 mL/min/1.73 m2,  is that "m2"?

Line 143: SC-based eGFR, "SCr"?

Typos:

Line 123:(CrCl) >130 mL/min/1.73 m2,  is that "m2"?

Line 143: SC-based eGFR, "SCr"?

Author Response

Thanks for your review, We have edited those typos in both lines 123,143

Reviewer 2 Report

The reviewer would like to thank the authors for answering the questions and remarks so accurately. 

However, the conclusion has not been amended in line with the changes in the main text. In particular, I think the conclusion should be more nuanced on the contribution of Bayesian PK with drugs other than vancomycin. Please reword. 

Author Response

Thanks for your review. We have amended the conclusion, deleted some sentences, and added sentences in lines 345, 346, and 347
